

# An enhanced integrated fuzzy logic-based deep learning techniques (EIFL-DL) for the recommendation system on industrial applications

Yasir Rafique[1], Jue Wu[1], Abdul Wahab Muzaffar[2] and Bilal Rafique[1]

[1] School of Computer Science and Technology, Southwest University of Science and Technology, Mianyang, China
[2] College of Computing and Informatics, Saudi Electronics University, Riyadh, Saudi Arabia

Corresponding authors
Jue Wu, wujue@aliyun.com
Abdul Wahab Muzaffar,
a.muzaffar@seu.edu.sa

## ABSTRACT

Industrial organizations are turning to recommender systems (RSs) to provide more personalized experiences to customers. This technology provides an efficient solution to the over-choice problem by quickly combing through large amounts of information and supplying recommendations that fit each user's individual preferences. It is quickly becoming an integral part of operations, as it yields successful and convenient results. This research provides an enhanced integrated fuzzy logic-based deep learning technique (EIFL-DL) for recent industrial challenges. Extracting useful insights and making appropriate suggestions in industrial settings is difficult due to the fast development of data. Traditional RSs often struggle to handle the complexity and uncertainty inherent in industrial data. To address these limitations, we propose an EIFL-DL framework that combines fuzzy logic and deep learning techniques to enhance recommendation accuracy and interpretability. The EIFL-DL framework leverages fuzzy logic to handle uncertainty and vagueness in industrial data. Fuzzy logic enables the modelling of imprecise and uncertain information, and the system is able to capture nuanced relationships and make more accurate recommendations. Deep learning techniques, on the other hand, excel at extracting complex patterns and features from large-scale data. By integrating fuzzy logic with deep learning, the EIFL-DL framework harnesses the strengths of both approaches to overcome the limitations of traditional RSs. The proposed framework consists of three main stages: data preprocessing, feature extraction, and recommendation generation. In the data preprocessing stage, industrial data is cleaned, normalized, and transformed into fuzzy sets to handle uncertainty. The feature extraction stage employs deep learning techniques, such as convolutional neural networks (CNNs) and recurrent neural networks (RNNs), to extract meaningful features from the preprocessed data. Finally, the recommendation generation stage utilizes fuzzy logic-based rules and a hybrid recommendation algorithm to generate accurate and interpretable recommendations for industrial applications.

# INTRODUCTION

In the era of big data, industrial applications generate vast amounts of data, which poses challenges in extracting valuable insights and providing accurate recommendations. Traditional recommendation systems (RSs) often struggle to handle the complexity and uncertainty inherent in industrial data (*Roozbahani et al., 2020*). To overcome these limitations, this article proposes an enhanced integrated fuzzy logic-based deep learning techniques (EIFL-DL) framework for RSs in industrial applications. RSs have become increasingly important in industrial applications, aiding users in making informed decisions and improving overall productivity. However, the complexity and uncertainty inherent in industrial data pose significant challenges for traditional RSs. To address these challenges, this article proposes the use of fuzzy logic-based deep learning techniques for RSs in industrial applications. Fuzzy logic is a mathematical framework that allows for the representation and handling of imprecise and uncertain information (*Bobadilla et al., 2013*). It provides a flexible and intuitive approach to modeling complex relationships and captures the vagueness often encountered in publication under the terms and conditions of the Creative Commons Attribution industrial data.

On the other hand, deep learning is a powerful machine learning (ML) technique that excels at extracting meaningful features and patterns from large-scale datasets. By integrating fuzzy logic with deep learning, we can leverage the strengths of both approaches to enhance the RSs in industrial applications (*Chen, Chen & Wang, 2015*). Fuzzy logic-based deep learning techniques allow for improved accuracy, interpretability, and handling of uncertainty in the recommendation process. The proposed approach involves several key steps. First, the industrial data is preprocessed to handle noise, outliers, and missing values (*Batmaz et al., 2019*). Fuzzy logic is then applied to represent the uncertainty and vagueness present in the data. The preprocessed data is fed into deep learning models, such as convolutional neural networks (CNNs) or recurrent neural networks (RNNs), to extract relevant features and patterns (*Dang, Moreno-García & Prieta, 2021*). These deep learning models are trained on fuzzy data representation to learn complex relationships and make accurate recommendations. This research proposes a customized RS that aims to create an intelligent decision-making support tool for customers in a cloud manufacturing system. However, one major disadvantage of knowledge-based RSs is the potential knowledge acquisition bottleneck (*Devipriya et al., 2020*). Explicitly defining recommendation knowledge can be time-consuming and labor-intensive. As the system relies on human-curated knowledge about products and their characteristics, keeping the knowledge base up-to-date and relevant may require ongoing efforts (*Nogueira dos Santos, Xiang & Zhou, 2015*).

The proposed RS utilizes neural networks for data regression, incorporates the analysis of past customer selections, and extracts key features from incoming manufacturing solution requests and available solutions (*Bai et al., 2017*). By employing this approach, the ML procedure can segment customers and generate a tailored recommendation list of manufacturing solutions based on their specific profiles (*Betru, Onana & Batchakui, 2017*). To validate the proposed system, a simulated case study was conducted within a cloud

manufacturing platform considered a large-scale manufacturing network with 950 suppliers and 60 customers. The evaluation includes training the regression learner with data from 15 customers and assessing the system's ML performance, industrial applicability, and user selection likelihood (*Catherine & Cohen, 2017*).

This study has made significant contributions to the field of industrial data processing and RSs. First, it has developed a new approach to integrating fuzzy logic into RSs to evaluate their effectiveness. Second, it has effectively determined the complexities and uncertainties of industrial data, to further enhance the performance of the recommender system. Last, the study employed the latest hybrid recommendation algorithm, intending to provide precise and easily understandable recommendations tailored to industrial applications. The rest of this article is structured as follows: In "Related Work", we provide a review of relevant research. "Materials and Methods" presents the introduction of our proposed RS, which is based on deep learning and tailored for handling the complexity and scale of industrial data. In "Experimentation Analysis", we outlined the evaluation process to demonstrate the system's performance. The obtained results are discussed, along with future work suggestions. Lastly, "Conclusion" offers the conclusions of this study.

## RELATED WORK

RSs have generated a lot of interest in industrial applications because they may aid users in making informed decisions and boosting productivity. To solve the challenges brought on by the complexity and ambiguity of industrial data, researchers have looked at the usage of fuzzy logic-based deep learning algorithms for RSs (*Chen et al., 2017a*). In this review of the literature, the significant studies on this subject are evaluated (*Chen et al., 2017b*). In this article, a fuzzy deep neural network (FDNN) method for commercial RSs is proposed. For handling ambiguity and removing pertinent characteristics, the FDNN blends fuzzy logic with deep learning (*Chen et al., 2017c*). The authors describe a deep learning model for industrial product recommendation that is based on fuzzy logic. To manage uncertainty and capture the ambiguous linkages between items and user preferences, fuzzy logic is applied. Cooperation Products are evaluated through filtering based on previous reviews from users, both explicit and implicit (*Chu & Tsai, 2017*). The strategy entails compiling a database of user preferences for different goods. It uses deep learning models to extract important features from industrial data, effectively manages uncertainty, and adds collaborative filtering to improve suggestion accuracy. In exploratory comparison experiments, the deep neural network performed better than the majority of previous techniques (*Dai et al., 2016a*), as a result, the authors suggest an improved fuzzy deep learning model for industrial process suggestion. A unique, multifaceted methodology for analyzing people's perceptions of medical services was presented by the proposed study. *Dai et al. (2016b)* evaluated the performance of basic and deep neural networks using textual, visual, and a combination of text and visual information. The deep neural network (*Dong et al., 2017*) outperformed the majority of other techniques in an exploratory comparison exercise. While deep learning approaches collect information linked to the industrial process, fuzzy logic is used to capture the ambiguity and uncertainty in the data. The researchers unveiled Auto Rec, a recommendation mechanism that anticipates

missing ratings by using Autoencoders. Using the appropriate ML techniques (*Hudson & Manning, 2018*) and the simulations offered by Apache Spark, it was able to extract valuable knowledge from enormous amounts of medical data. In their trials, *Lee et al. (2016)* showed that, for the Movie Lens and Netflix datasets, Auto Rec surpassed matrix factorization and restricted Boltzmann machine (RBM)-based collaborative filtering in terms of accuracy. In this study, a neural network was integrated into an optimization-based methodology (*Lei et al., 2016*).

## Category systems for RSs in industrial applications

RSs play a crucial role in various industrial applications by helping users discover relevant products, services, or content based on their preferences and behaviors. Here are some categories of RSs commonly used in industrial applications:

### Collaborative filtering recommendation

Collaborative filtering (CF) is a popular approach used in RSs to evaluate products based on users' historical ratings (explicit or implicit) from the data. The technique builds a database of user preferences for items and then identifies active users' neighbors who share similar purchase preferences. CF (*Beel et al., 2016*) can be categorized into two main types: item-based filtering and user-based filtering. In user-based CF, the method involves two key stages to predict ratings for a specific user. First, it identifies similar users to the target user based on their historical ratings and preferences. Next, it obtains ratings from similar users and utilizes them to generate personalized recommendations for the active user. Various similarity measures have been proposed in the literature to calculate the similarity among users. Some commonly used measures (*Beel & Langer, 2015*) include mean-squared difference, Pearson correlation, cosine similarity, Spearman correlation, and adjusted cosine similarity. CF is widely adopted in RSs due to its effectiveness and ability to automatically learn embedding. The term "embedding" refers to mapping items to a sequence of numbers or vectors. By representing items with learned vectors, algorithms can discover relationships between items and extract their features automatically. Overall, CF (*Beel et al., 2013*) is a powerful recommendation technique that does not require domain knowledge, as it can learn from user-item interactions and provide accurate recommendations based on user preferences.

### Content-based recommendation

Content-based approaches in RSs aim to create a user profile based on their historical interactions to predict ratings for unseen items. Successful content-based methods make use of tags and keywords associated with items to build item profiles and understand user preferences. The utility of content-based filtering (*Bellogín & Said, 2021*) is often measured using heuristic functions, with the cosine similarity metric being a common choice. Content-based filtering is suitable in cases where item features can be easily extracted and represented. However, it may not be practical when features (*Berg, Kipf & Welling, 2017*) need to be manually entered, especially in scenarios with a large number of new products being added regularly. One advantage of content-based filtering is that it does not rely on other users' data, making it more scalable for handling many users. Predicted

recommendations are user-specific, which allows the system to handle a diverse set of users effectively. Unlike CF, content-based filtering does not face cold-start issues (*Beutel et al., 2019*). It can suggest new items to users even before a substantial number of users have rated those items. However, content-based filtering has some limitations. If the available content does not provide enough information to precisely differentiate products, the recommendations may not be accurate. The effectiveness of content-based filtering heavily relies on the quality and richness of the item features used to represent the items (*Biega, Gummadi & Weikum, 2018*).

### Demographic-based recommendation

According to various quantitative research articles, CF techniques can benefit from the incorporation of demographic correlation (*Bogers et al., 2020*). Demographic RSs operate by categorizing users based on their demographic attributes, such as age, gender, and language, to generate personalized recommendations. These RSs are particularly valuable when there is limited product information available. Demographic RSs offer solutions to two critical issues in RSs: scalability and cold-start problems. By using user attributes (*Bonhard et al., 2006*) as demographic data, these systems can provide recommendations without relying on user ratings, which is a key advantage over content-based and collaborative filtering techniques. One of the key benefits (*Bostandjiev, O'Donovan & Höllerer, 2012*) of demographic filtering RSs is their speed and simplicity in generating recommendations with only a few observations. They can quickly provide relevant suggestions based on users' demographic information (*Bosteels, Pampalk & Kerre, 2009*).

### Utility-based recommendation

Utility-based RSs provide recommendations by creating a utility model for each item personalized to the user. In this approach (*CarlKadie, 1998*), the system builds utility functions that consider multiple attributes of users and items to determine the utility of each item for the user explicitly. One of the key advantages (*Burke, 2002*) of utility-based RSs is their ability to factor in non-product attributes when calculating utility functions. This means that attributes such as product availability, vendor reliability, and other relevant factors can be considered, making the recommendations more comprehensive and informative. Utility-based RSs are particularly useful for scenarios where real-time information about items is critical. By computing the utility explicitly (*Burke, 2010*), the system can check real-time inventory and consider the current features of an item, providing users with up-to-date information and visualization of the item's status. Unlike some other recommendation techniques (*Burke et al., 2016*) that rely on long-term user profiles, utility-based systems focus on evaluating recommendations based on the user's current needs and the available options. This allows the system to adapt quickly to changing user preferences and evolving item attributes. Overall, utility-based RSs (*Cañamares, Castells & Moffat, 2020*) offer a flexible and dynamic approach to recommendations by incorporating various attributes and real-time information. This enhances the relevance and accuracy of the recommendations, making them more useful for users with varying preferences and requirements (*Campbell, 2006*).

### Knowledge-based recommendation

Knowledge-based RSs utilize explicit knowledge about products and users to create a knowledge-based criterion for generating recommendations (*Castells, Hurley & Vargas, 2021*). Unlike other RS approaches that rely on user ratings or historical data, knowledge-based RSs do not require a large initial dataset. *Celik et al. (2018)* recommendations are independent of user ratings, and they focus on evaluating products that match the user's specific needs and preferences. One of the main advantages of knowledge-based RSs is their ability to avoid the typical ramp-up problem often associated with ML-based recommendation approaches. In traditional RSs (*Celma & Herrera, 2008*) that rely on user ratings, the system may struggle to make accurate recommendations until the user has rated a substantial number of items. Knowledge-based systems bypass this issue since their recommendations are not dependent on a user's rating history. Another benefit is that knowledge-based RSs do not need to collect user-specific information or create user profiles, as their recommendations are independent of individual user tastes (*Celma, 2010*). This simplifies the system's implementation and removes potential privacy concerns associated with collecting and storing user data. Due to these advantages (*Chalmers et al., 1981*), knowledge-based RSs can serve as valuable stand-alone systems, particularly in situations where there are limited user data or when it is challenging to gather explicit user preferences. Additionally, they can complement other types of RSs by providing targeted recommendations based on specific knowledge about products and users (*Chao & Lam, 2011*).

### Hybrid-based recommendation

Hybrid systems in RSs combine two or more recommendation techniques to achieve improved performance. The primary objective of hybrid systems is to address the limitations and drawbacks of individual recommendation approaches. By leveraging the strengths of different techniques (*Chen, Shih & Lee, 2016*), hybrid systems aim to provide more accurate and diverse recommendations to users.

# MATERIALS AND METHODS

## System overview

The RS plays a vital role in industrial applications, aiding users in making informed decisions and enhancing overall productivity. To address the challenges posed by the complexity and scale of industrial data, this article proposes a deep learning-based approach for RSs in industrial applications as illustrated in Fig. 1. The proposed framework is developed to address three key challenges:

1. Generating data cleaning techniques to serve as enhanced input for the recommenders.
2. Improving the accuracy of extracting spatial and local features product recommendations.
3. Improving the deep learning model. The primary objective of this framework is to enhance the overall performance, with a specific focus on improving accuracy RSs.

An industrial RSs, implemented with reinforcement learning (RL) was devised to deliver personalized suggestions or advice to users, tailored to their preferences and real-time conditions. This versatile system can be effectively employed across multiple sectors,

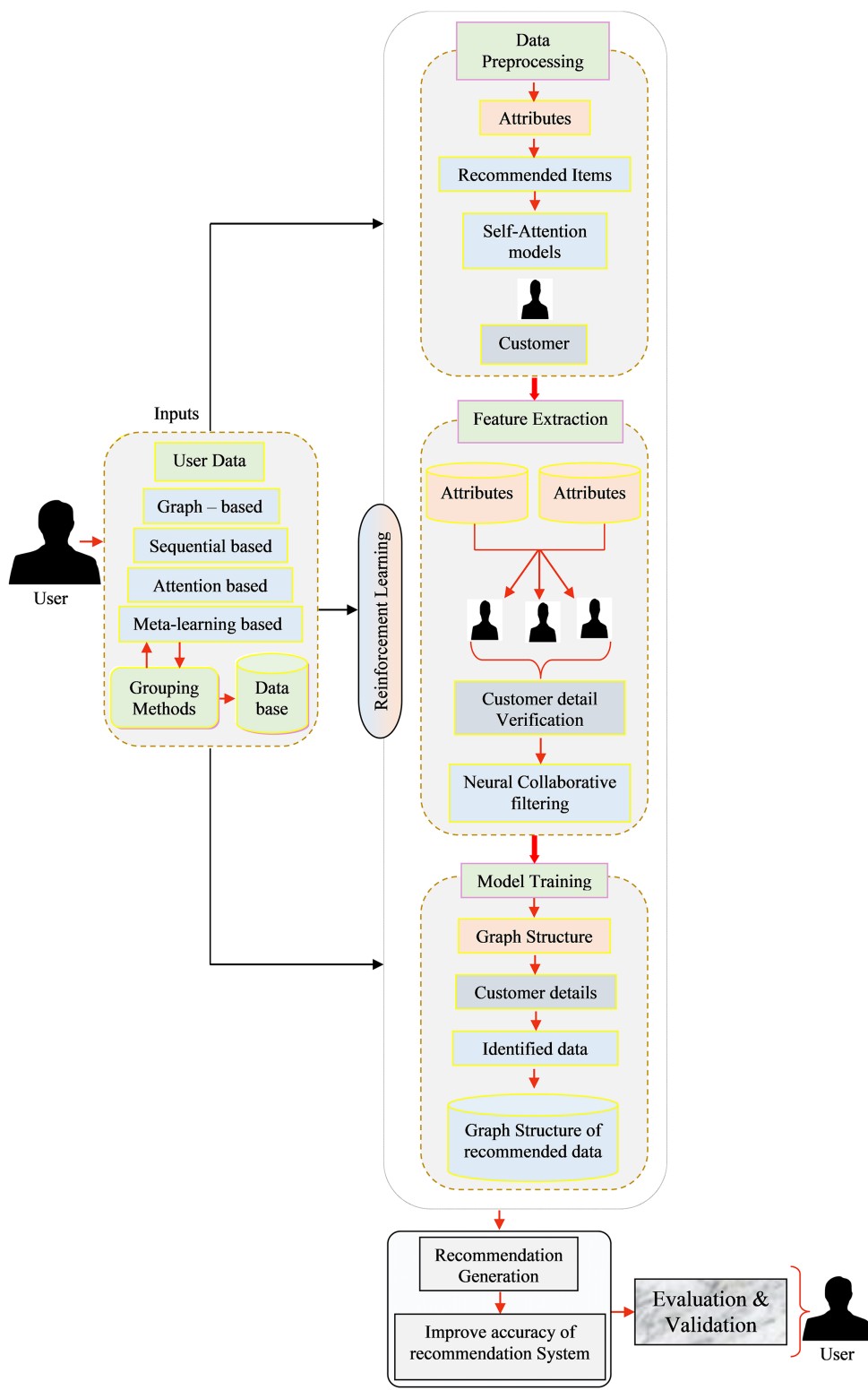

**Figure 1** Overall view of the proposed recommendation systems framework.

**Table 1 Description of the symbols.**

| Notation | Description |
| --- | --- |
| $D$ | Data cleaning |
| $P$ | Preprocessing threshold |
| $D_i^s$ | Instance of the dataset being evaluated |
| $y$ | Value selected from the dataset for extraction |
| $x(n)$ | Input feature vector at iteration $n$ |
| $\alpha$ | Learning rate or weight adjustment factor |
| $d_j^n$ | Delta value associated with the $j$-th parameter at iteration $n$ |
| $R$ | Recommended model output |
| $\phi$ | Activation function or transformation applied in the model |
| $R_i^G$ | Generated recommendations for item $i$ |
| $\lambda_i$ | Input embeddings for item $i$ |
| $\delta$ | Accuracy metric |
| $\gamma$ | Weighting factor |

including manufacturing and supply chain management. Below, we outline the key components involved in the development of such a system and the description of the main notations has been illustrated in Table 1:

1. User inputs

    (a) User preferences: User choices and historical interactions play a pivotal role in the recommendation process, collected either through explicit feedback like ratings and reviews or implicit feedback such as purchase history, click-through rates, and sensor data.

    (b) Contextual information: In industrial settings, contextual factors, like the user's role, Industrial specific constraints, and real-time data (*e.g.*, inventory levels and machine status), are indispensable for generating relevant recommendations.

    (c) Interaction data: The system must amass data on user interactions within the environment or system, encompassing actions taken and their outcomes.

2. Data preprocessing

    (a) Aggregate data from various sources, including user interactions, product details, and contextual information.

    (b) Data cleaning: Refine and sanitize the data, addressing missing values, outliers, and inconsistencies.

    (c) Data transformation: Convert raw data into a suitable format for further analysis, including the encoding of categorical variables and normalization of numerical features.

    (d) Temporal data handling: In an industrial context, time-series data often holds significance for comprehending user behavior.

(e) Appropriate treatment of temporal aspects is vital.

3. Feature extraction

(a) Feature engineering: Construct relevant features from the preprocessed data, potentially incorporating attributes related to product specifications, user roles, historical actions, and environmental conditions.

(b) Embedding: Utilize embedding or feature representations to effectively capture the relation ships between users, products, and contextual elements.

4. Model training

(a) RL: RL models are well-suited for RS due to their ability to learn through interactions with the environment and optimising for long-term rewards.

(b) RL environment: Define the RS as an RL environment, where users act as agents, making recommendations and receiving rewards based on user feedback.

### Data preprocessing

In the initial phase, text preprocessing is applied to the text reviews in dataset A and various preprocessing tasks are performed, resulting in a vector comprising unique integer indexes (IDs). This vector serves as the input for word embedding as shown in Eq. (1). The process converts IDs into dense real-valued vectors with semantic meaning. The word embedding ensures that words with similar meanings have similar vectors and are positioned closely in a high-dimensional semantic space. Subsequently, the real-valued vector of each review, along with its explicit rating, becomes the input for training the sentiment predictor. In this stage, the industrial data is preprocessed to handle noise, outliers, and missing values. Data cleaning techniques are applied to ensure the quality of the data. The data may also undergo normalization or standardization to bring it to a consistent scale.

$$|(D_i^s - D_i^s)| <= P \tag{1}$$

where $D$ = Data cleaning, $P$ = Preprocessing, Eq. (1) represents an absolute difference between the two instances, $D_i^s$ indicating that the absolute value of their discrepancy should be less than or equal to a defined threshold, denoted as $P$. This mathematical expression sets a condition ensuring that the dissimilarity between the two instances remains within a specified limit $P$.

### Feature extraction

In the age of information overload, RSs have become essential in mitigating this issue and are extensively utilized across various online services, such as E-commerce, streaming platforms, and social media sites. CF with implicit feedback is a core component of these RSs. Among the variations of CF, matrix factorization is the most commonly used approach, where a fixed inner product of the user-item matrix is utilized to capture user-item interactions. A more recent advancement (*Chen et al., 2017c*) performance in this domain is neural CF (NCF), which replaces the traditional user-item inner product

with a neural architecture. This innovative approach incorporates neural networks to model and enhance user-item interactions, providing a more effective and dynamic recommendation mechanism. Equations (1) and (2) shows that there is a need to develop a more specialized and improved extraction function dedicated to modelling the latent feature interaction between users and items. NCF emerges as a solution to address this issue by,

$$Extraction = \frac{\left|\left\{j : \left|\left(D_i^s - D_i^s\right)\right| \le P\right\}\right|}{\left|\left\{j : j \varepsilon dataset\right\}\right|} * P\left(\frac{dy}{dx}\right) \tag{2}$$

Equation (2) describes a data extraction process where the value of $y$ is selected from a dataset using the conditions $D_i^s - D_i^s$ with a limit $P$.

### Model training

Once the features are extracted, the deep learning model is trained using appropriate algorithms such as back propagation or gradient descent. The model is optimized to minimize the loss function and maximize the accuracy of the RS. The delta rule for training simple two-layer networks has been previously explained. However, the challenge lies in adapting weights in the hidden layer(s) of the multilayer perceptron (MLP), or more precisely, in calculating the required adjustments for the hidden units. The training process (*Chen, Chou & Kauffman, 2009*) Eq. (3) involves feeding the extracted features along with the corresponding target labels. The backpropagation algorithm is an extension of the delta rule and relies on gradient descent to minimize the sum squared difference between the target and actual network outputs. In the training model, where t represents training and a represents accuracy. Typically, the analysis assumes the presence of a semi-linear activation function, such as the sigmoid, which is both differentiable and monotonic.

$$t_i = \alpha * \alpha_j^n + (1 + \alpha) * d_j^n * x(n). \tag{3}$$

### Recommendation generation

After training the model, it generates recommendations based on user input or specific queries. The deep learning model takes the input data and processes it through the trained network to produce recommendations. where R represents the recommended model and $\phi$ represents personalized output. The recommendations may be ranked based on relevance or personalized to individual user preferences.

$$R_i^G = \phi_{(out)}(\phi(y)(\ldots(\phi_1(\phi_2(\phi_{(length)}(\phi_{(embedding)}(\lambda_i)))))\ldots)) \tag{4}$$

Several techniques can be applied to the training data to effectively enhance an RSs performance:

1. Excluding popular items from the training data, especially when users can easily discover them on their own. This approach is beneficial as it helps avoid redundant recommendations that users might not find it useful.

2. Scaling item ratings based on the user's value, such as their average transaction value. By doing this, the model can better learn to recommend items that are likely to appeal to loyal or high-value customers, leading to more personalized and relevant recommendations.

### Evaluation and validation

The performance of the RS is evaluated using appropriate evaluation metrics, such as precision, recall, or means average precision. Validation techniques, such as cross-validation or hold-out validation may be employed to assess the generalization capability of the model. The system's performance is compared with baseline approaches or existing RSs to demonstrate its effectiveness in the following equations.

$$F = \frac{(2 * precision * recall)}{precision + recall} * h(x) \log \int_0^h x^3 \tag{5}$$

$$precision = \frac{Corresponding\ item}{total\ no.\ of\ items} \tag{6}$$

$$recall = \frac{Corresponding\ item}{total\ no.\ of\ items} \tag{7}$$

The integration term involves the definite integral of $x^3 \log(x)$ concerning $x$ from 0 to $h$. The F1 score combines precision and recall into a single value, giving a harmonic mean that balances the trade-off between these two metrics in classification performance assessment. By leveraging deep learning techniques, the proposed approach aims to improve the accuracy and scalability of recommendation systems in industrial applications. The use of deep learning models (*Chen & Konstan, 2015*) allows for the extraction of intricate patterns and representations from large-scale industrial data, leading to more precise and relevant recommendations. Additionally, the proposed approach can adapt to dynamic and evolving industrial environments by continuously updating the recommendation model using new data.

$$extraction = \frac{Corresponding\ item + nonCorresponding\ item}{N} \tag{8}$$

In conclusion, the proposed deep learning-based RS for industrial applications offers a promising solution to handle the challenges posed by complex and voluminous industrial data. The overview presented above provides a high-level understanding of the proposed approach (*Christakopoulou, Radlinski & Hofmann, 2016*), and the accuracy measured in Eq. (9) suggest that further research and experimentation are required to validate its effectiveness and optimize its performance in various industrial settings.

$$\delta = accuracy_\infty \sum_{i=1}^p \frac{\exp(\gamma . \sin(y_a, y_{ail}))}{\sum X \in \gamma^{da} \exp(\gamma . \sin(y_a, f_a(x)^0))} \tag{9}$$

In Eq. (9), $\gamma$ functions as a weighting factor that affects the accuracy metric $\delta$. It is essential for equilibrating the contributions of various components within the system. Specifically, $\gamma$ can be construed as a parameter that modulates the impact of diverse predictions or inputs (*e.g.*, $y_a$, $y_{all}$) on the ultimate accuracy evaluation.

This weighting aims to refine the recommendation model's sensitivity to the intricacies of extensive industrial data, hence improving the accuracy and importance of the generated recommendations. Incorporating $\gamma$ into the equation enables the model to dynamically adjust to changes in the data landscape, hence maintaining its efficacy in changing industrial contexts.

Data preprocessing is a crucial step in the deep learning pipeline, as it involves cleaning, transforming, and preparing the data to ensure its quality and compatibility with deep learning models. To enhance the efficiency and effectiveness of data preprocessing, this article proposes a deep learning-based data preprocessing measured quality inspector for industrial applications (Eq. (10)). The inspector leverages deep learning techniques (*Christakopoulou, Radlinski & Hofmann, 2016*) to automate and optimize the data preprocessing process, ensuring high-quality data for subsequent analysis and modelling tasks.

$$t = (x_i + x_j)$$

where, $x_{ij} = \frac{at(A_i \| A_j)}{at(A_i \| A_j)} * x_{ij} = \frac{1}{x_{ij}}$

The weight attributes normalized represented as $(w_0, w_1, \ldots, w_n)$ as follows:

$$w_i = \frac{w_i + w_j}{\sum_{i=0}^{\alpha} (w_i + w_j)^n} \tag{10}$$

The equation represents the normalization of weight attributes in a vector $(w_0, w_1, \ldots, w_n)$. Each weight $w_i$ is updated to be the sum of $w_i$ and $w_j$ divided by the total sum of all weights raised to the power of n. This normalization process ensures that the weights collectively sum to 1, effectively distributing the influence of each weight in the vector proportionally. The parameter $\alpha$ denotes the upper limit of the summation, and the formula is designed to adjust weights to maintain relative proportions while ensuring the sum of weights remains constant.

In the initial phase, the text reviews from dataset A undergo preprocessing, resulting in a vector containing unique integer indexes (IDs). This vector serves as input for a word embedding process, where the IDs are transformed into dense real-valued vectors with semantic meaning. This transformation allows words with similar meanings to have similar vectors, effectively positioning them close to one another in a high-dimensional semantic space. Subsequently, the real-valued vectors representing each review and its explicit rating are used as inputs to train the sentiment predictor. This predictor analyses and predicts the sentiment or emotion conveyed in the text reviews. The generative adversarial networks (GAN) training process converges when the generator becomes proficient enough to produce data that the discriminator can hardly differentiate from the real data. At this point, the GAN in Fig. 2 reaches Nash equilibrium and the generator has

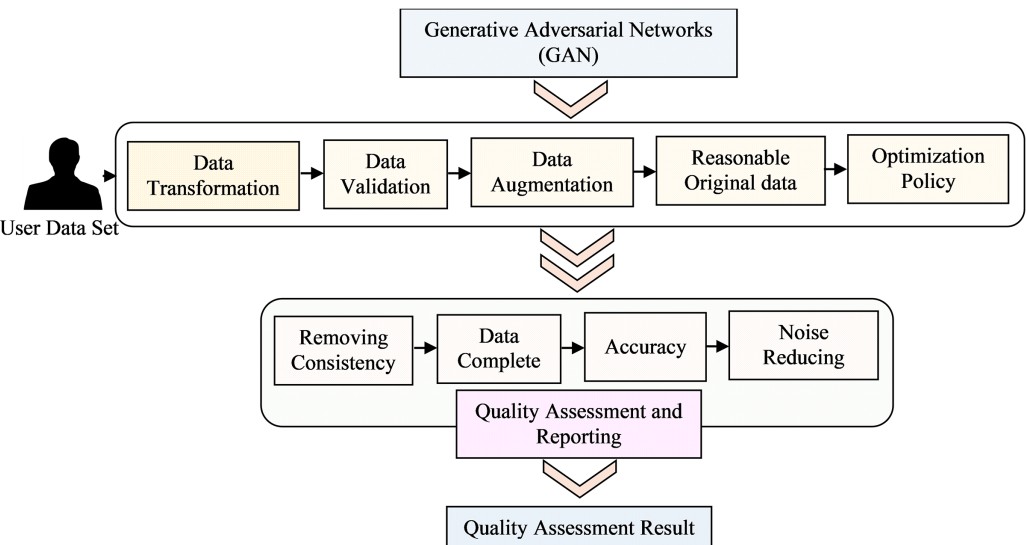

**Figure 2 Architecture for data preprocessing quality assessment model deep learning techniques.**

effectively learned to model the data distribution of the training set. RSs can be evaluated using different types of data, such as explicit feedback (*e.g.*, ratings or likes), implicit feedback (*e.g.*, browsing behaviour or purchase history), or a combination of both. Each data type presents unique challenges and requires specific evaluation techniques to extract meaningful insights. Data sparsity refers to the situation where only a small fraction of potential user-item interactions are observed in the data. In many real-world scenarios, users interact with only a small subset of available items, resulting in sparse data matrices. Addressing data sparsity is crucial for accurate evaluation (*Jalili et al., 2018*) and efficient recommendation algorithms. The cold-start problem occurs when new users or items have limited or no historical interactions in the data. Evaluating RSs under cold-start conditions is challenging as the system needs to provide meaningful recommendations despite the lack of historical data for new users or items. Item metadata comprises various attributes that describe the items, such as genre, category, author, or producer. By considering item attributes, RSs can suggest items that align more closely with user preferences and improve recommendations for less-known items.

### Qualitative and quantitative dataset

In the evaluation of RSs, data collection can encompass both quantitative and qualitative methods. Quantitative data collection relies on highly structured instruments, such as standardized surveys or questionnaires, to gather user feedback directly. These structured approaches facilitate the validity and comparability across studies and enable deductive analysis using statistical methods, allowing for generalization to the wider population. On the other hand, qualitative data collection methods, such as interviews, focus groups, and participant observations, are used to gain a deeper understanding of the study's sample. Qualitative data are collected in the form of notes, videos, audio recordings, images, or text

documents, offering valuable insights into users' preferences, perceptions, and experiences with the RS. Combining quantitative and qualitative data allows for a comprehensive evaluation, enabling researchers to triangulate findings and derive more nuanced and context-specific conclusions about the system's performance and user interactions.

**Data collection:**

Data collection methods in the evaluation of RSs can be categorized based on their focus on contemporary or historical events. Historical data collection methods rely on past events, utilizing existing datasets or data retrieved from social media platforms to assess system performance. In contrast, contemporary data collection methods investigate current user interactions and preferences. Another important distinction in evaluation methods is user involvement. Offline studies do not require active user participation with the recommender system, as they solely rely on historical data for analysis. On the other hand, user-centric evaluations (*Cosley et al., 2003*) involve direct user engagement, where users interact with the recommender system, provide feedback, or participate in user studies. While offline studies are often more efficient and less resource-intensive, user-centric evaluations offer valuable insights into user experiences and preferences, making them essential for understanding real-world user interactions with the recommender system. However, user-centric evaluations can be more expensive in terms of time and money due to the need for user recruitment, data collection, and user feedback gathering. Both types of evaluations play a crucial role in providing a comprehensive understanding of the recommender system's performance and user satisfaction.

**Data cleaning**

The data cleaning process involves implementing three filters and one preprocessing task. A final dataset containing 2.90 million rows and five features is obtained. To facilitate the two-phase experiments, 600,00 records are randomly selected from the final dataset, with 120,00 rows corresponding to each rating label. This selection is used to create the dataset for the experiments. After the preliminary set, a final dataset is obtained, consisting of 2,216,008 rows with five features. Among these data records, only 46,954 have a rating equal to 1. To create dataset B, a total of 235,000 records are randomly selected, with 46,000 rows corresponding to each rating label. The two most effective data-cleaning techniques were then implemented to reach at this final dataset. The inspector utilizes deep learning models, such as autoencoders or (GANs), to identify and remove noisy or erroneous data points. These models can learn the underlying patterns and structures of the data, allowing them to detect and eliminate outliers, missing values, or inconsistent entries.

**1) Autoencoders**

An autoencoder is a neural network designed to acquire efficient data representations, primarily for dimensionality reduction or noise reduction. The design comprises two primary components: an encoder that compresses the input data into a latent space

representation and a decoder that reconstructs the original data from this compressed form. The loss function for an autoencoder is often the mean squared error (MSE) between the input data and its reconstruction, expressed as:

$$L_{AE} = \frac{1}{N} \sum_{i=1}^{N} ||x_i - \hat{x}_i||^2$$

where $x_i$ denotes the original input data, $\hat{x}_i$ signifies the reconstructed output, and $N$ indicates the number of data points. The autoencoder is trained by backpropagation to reduce the reconstruction loss. Throughout training, the model acquires the ability to discern the fundamental patterns within the data, efficiently eliminating additional noise. The resulting latent representation may subsequently be utilized for downstream activities (*Kuchaiev & Ginsburg, 2018*).

**2) Generative adversarial networks**

Generative adversarial networks (GANs) comprise two adversarial neural networks: a generator that endeavors to produce authentic data samples and a discriminator that assesses the authenticity of the input data, determining whether it is real or fabricated. This adversarial process results in the generator creating high-quality data samples, which can assist in noise reduction. The loss function for GANs encompasses both the generator and the discriminator (*Su et al., 2022*).

*Discriminator:*

$$L_G = -\mathbb{E}_{z \sim P_z}[\log D(G(z))]$$

*Generator:*

$$L_D = -\mathbb{E}_{x \sim P_{data}}[\log D(x)] - \mathbb{E}_{z \sim P_z}[\log(1 - D(G(z)))]$$

In this context, $D(x)$ represents the discriminator's probability assessment that $x$ is an authentic sample, $G(z)$ denotes the generated sample derived from random noise $z$, while $P_{data}$ and $P_z$ signify the true data distribution and the noise distribution, respectively. The generator and discriminator are changed iteratively throughout the training process. The discriminator is trained to differentiate authentic data from counterfeit data, whereas the generator is trained to deceive the discriminator into categorizing its output as authentic. This process persists until the generator yields data samples that closely mimic the authentic data, thereby diminishing noise and enhancing the overall quality of the dataset.

**Data transformation**

Deep learning models, such as vibrational auto encoders (VAEs) or self-organizing maps (SOMs), can be employed to transform the data into a more suitable representation. These models learn the latent representations of the data, capturing its essential features

and reducing its dimensionality. This transformation can enhance the efficiency and effectiveness of subsequent analysis tasks.

**Data validation**

The inspector employs deep learning models, such as RNNs or long short-term memory (LSTM) networks, to validate the integrity and consistency of the data. By using the forward states of the LSTM, it can efficiently capture information from past features in the sequence, allowing it to understand the context leading up to the current position. Conversely, by leveraging the backward states of the LSTM, it can effectively utilize future features in the sequence, providing insight into the context that follows the current position. This combination of information from both ends of the sequence allows the LSTM to make more informed estimations of the output and better capture long-term dependencies and patterns within the data. This bidirectional processing can be particularly beneficial in tasks where understanding the complete context of the sequence is essential for accurate classification, prediction, or generation. These models can learn the temporal dependencies and patterns in the data, allowing them to identify any inconsistencies or anomalies that may affect the reliability of the data.

**Data augmentation**

Deep learning techniques, such as generative models or adversarial networks, can be utilized to augment the data by generating synthetic samples that closely resemble the original data. Introducing synthetic user-item interactions by randomly pairing users and items, or by duplicating existing interactions with slight modifications Item features: Modifying the item features to create new representations or augment the existing ones. For example, generating image variations, textual paraphrasing, or adding noise to the item descriptions. The item features (*Cremonesi et al., 2011*) modify the item features to create new representations or augment the existing ones. For example, generate image variations, paraphrase text, or add noise to the item descriptions. The user features the user features by incorporating additional information or generating variations based on user characteristics. This augmentation can help overcome issues related to limited data availability and improve the robustness and generalization capability of the subsequent deep-learning models.

**Quality assessment and reporting:**

The inspector evaluates the quality of the preprocessed data using appropriate metrics, such as data completeness, consistency, or accuracy. It generates comprehensive reports summarizing the quality assessment results, allowing stakeholders to gain insights into the data quality and make informed decisions for further analysis or modelling tasks. The first phase of experiments are conducted in two steps. In the initial step, the objective is to determine the optimal parameters for each model to achieve the best rating prediction results. This is accomplished by conducting various rapid experiments using a smaller version of dataset *A*. Once the parameters are identified, and the best-performing version

of each proposed algorithm is defined, the second step is executed. In this step, the complete dataset $A$ is utilized to test the quality inspectors. The $Q_t$ mainly used the quality of the performance and $V(P_n)$ in mean by average performance. Notably, all three tasks are described and run for both steps.

$$Q_t = m_t(s) = \sum_{k=1}^{0} Performaer(P_n) = 0.0731 \tag{11}$$

and

$$PerformaeV(P_n) = 1 - m(t) = 0.0643$$

By leveraging deep learning techniques, the proposed data preprocessing quality inspector aims to automate and optimize the data preprocessing process in industrial applications. It offers advantages such as improved efficiency, enhanced data quality, and reduced manual effort. The inspector enables organizations to leverage the power of deep learning to ensure high-quality data, leading to more accurate and reliable results in subsequent analysis and modelling tasks. The deep learning-based data preprocessing quality inspector presents a promising approach for enhancing the data preprocessing stage in industrial applications. The integration of deep learning models allows for automated and optimized data preprocessing, ensuring high-quality data for downstream tasks (*Cremonesi et al., 2008*). Further research and experimentation are required to validate the effectiveness and performance of the proposed inspector in various industrial settings. The evaluation of the different quality inspector models is based on indicators accuracy and the total time taken to run each model. Accuracy is computed when $k = 0$, and accuracy 1 when $k = 1$, using Eq. (2) where k represents the allowed prediction threshold. The third indicator is modelled as follows:

$$Q_t(N) = \sum_{K=i=0..n}^{0} P_i Q(S_i) = 0.37 \tag{12}$$

and

$$PerformaeP(n) = 1 - 0.37 = 0.63$$

**Evaluation metrics**

The evaluation of recommendation algorithms involves considering a wide range of facets to assess their performance thoroughly. As a result, the evaluation of RSs relies on a diverse set of metrics, which can be utilized for different experiment types. Most of the presented metrics have been developed for offline experiments, which are dominant in the field. These metrics encompass various aspects of evaluation, including predictive accuracy, classification accuracy, rank accuracy, prediction-rating correlation, utility optimization, coverage, novelty, diversity, serendipity, confidence, and learning rate. Different researchers have classified the available metrics from different perspectives, such as classification metrics, predictive metrics, and coverage metrics. and more. The

**Table 2 Metrics encompass various aspects of evaluation protocols.**

| Metric | Proposed model | Mean absolute error | Mean squared error | Mean reciprocal rank | Normalized discounted cumulative gain |
|---|---|---|---|---|---|
| Predictive accuracy | 0.5 | 0.7 | 0.09 | 0.13 | 0.19 |
| Classification accuracy | 0.16 | 0.34 | 0.31 | 0.23 | 0.21 |
| Rank accuracy | 1 | 0 | 1 | 1 | 0 |
| Prediction-rating correlation | Positive | Not positive | Not positive | Positive | Not positive |
| Utility optimization | 0.012 | 0.021 | 0.029 | 0.031 | 0.028 |
| Coverage | 87% | 76% | 79% | 82% | 86% |
| Novelty | 97% | 91% | 91.3% | 95.2% | 94.5% |
| Diversity | 0.62 | 0.71 | 0.56 | 0.62 | 0.67 |
| Serendipity | 0.70 | 0.84 | 0.77 | 0.72 | 0.80 |
| Confidence | 0.63 | 0.90 | 0.81 | 0.64 | 0.80 |
| Learning rate | 0.56 | 0.71 | 0.72 | 0.81 | 0.68 |

comprehensive set of evaluation metrics provided in Table 2, ensures that the performance of recommendation algorithms is rigorously assessed from multiple dimensions, aligning with the diverse requirements of the recommendation tasks and user satisfaction.

The main objective is to consolidate and systematically organize the existing knowledge related to predictive performance metrics for recommendation algorithms. By bringing together and structuring this knowledge, our work will provide a comprehensive and coherent understanding of the various evaluation metrics used to assess the predictive capabilities of RSs. This consolidation will help researchers and practitioners to better navigate and choose appropriate evaluation metrics based on their specific evaluation objectives and the characteristics of their recommendation algorithms. Ultimately, our work aims to enhance the transparency and comparability of evaluation results, contributing to the advancement of RS research and development.

### Deep learning-based recommendation techniques

Deep learning-based recommendation techniques refer to the application of deep learning models and algorithms for generating personalized recommendations to users. These techniques leverage the power of deep neural networks to extract complex patterns and representations from large-scale datasets, enabling more accurate and effective RSs. In recommendation filtering the prediction for a user in structured and non-structured scenarios is given by Eq. (13).

$$\hat{d}_{ni} = f(x^t . n_u^{coff} . n_i^{item} | x, y, \theta) \tag{13}$$

$$\lambda = \sum_{k=i,\ldots..n} x(P) + \hat{r}_{ui} + (1 - \hat{r}_{ui}) \log (1 - \hat{r}_{ui})^2. \tag{14}$$

The equation represents a loss function, denoted as $L$, commonly used in machine learning (ML), particularly in the context of collaborative filtering (CF) or recommender

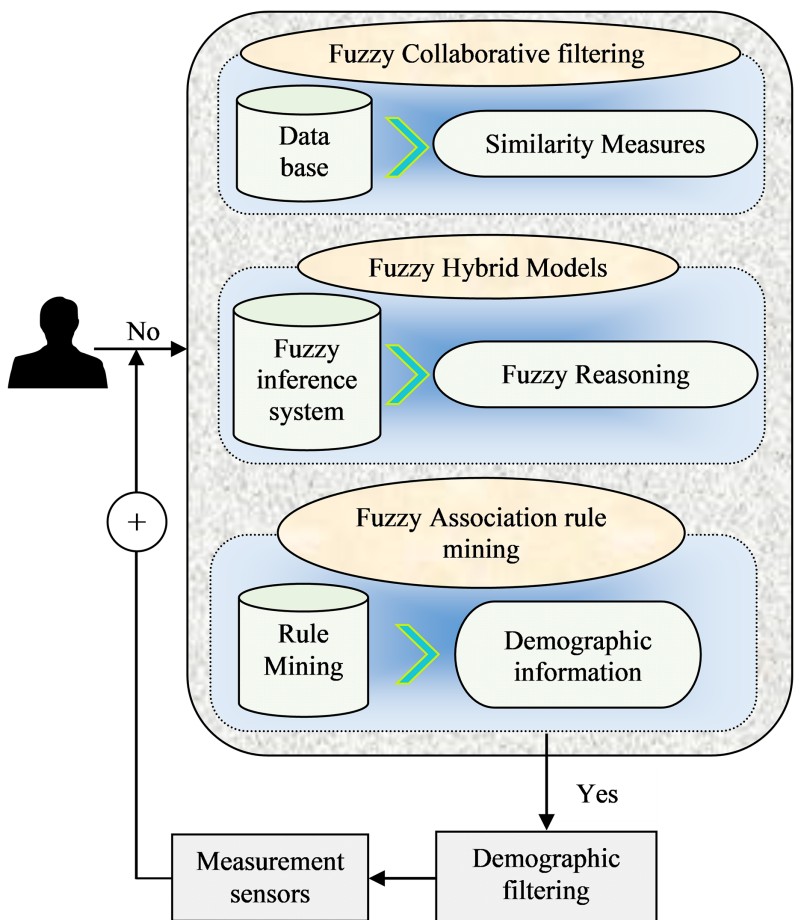

**Figure 3** **Analytical framework of fuzzy Logic-based recommendation model.**

systems (RSs). The summation, denoted by $\sum$, iterates over a range from index $i$ to $n$. The terms in the summation involve the prediction error for a specific user-item pair $(i, k)$, where $x(P)$ represents the predicted rating for the user-item pair, $r_{ui}$ is the actual rating, and $(1 - r_{ui})$ is the complement of the actual rating. The loss is calculated using the logarithmic square of the difference between the actual rating $r_{ui}$ and the predicted rating $x(P)$. The logarithmic term, $\log(1 - r_{ui})$, is squared to amplify the impact of larger errors. Deep learning models are designed to learn hierarchical representations of data, capturing intricate relationships and patterns (*Creswell & Creswell, 2005*). They can automatically discover and extract relevant features from raw input data, which is particularly beneficial for RSs as they often deal with high-dimensional and unstructured data, such as user-item interactions, textual descriptions, or images.

## Integrated fuzzy logic-based recommendation techniques

Figure 3 shows the integrated fuzzy logic-based recommendation techniques that utilize fuzzy logic principles and methods in RSs to handle uncertainty and imprecision. Here is an example of a mathematical model that combines fuzzy logic with CF:

### Fuzzy CF model

Fuzzy set theory encompasses mathematical approaches that are particularly adaptable and suitable for dealing with situations involving incomplete information, imprecise classifications, and gradual preferences. This theory and logic offer a means to quantify uncertainty arising from vagueness and imprecision. At the core of fuzzy set theory are membership functions, which form the basis of fuzzy sets. These functions possess a possibilistic interpretation, assuming the presence of a property and assessing its strength concerning other members within the set. A fuzzy set $A$ in the domain space $X$, denoted as $\mu_A : X \rightarrow [0,1]$, is characterized by its membership function, where $X$ represents the universe of discourse. Alternatively, a fuzzy set $A$ can be characterized using a set of pairs.

Let's consider a recommendation system with a set of users $U = \{u_1, u_2, u_3, \ldots, u_n\}$ and a set of items $I = \{i_1, i_2, i_3, \ldots, i_m\}$. The goal is to generate recommendations for a target user $u_t$ based on the preferences of similar users. The interpretation of the fuzzy membership function $\mu_A(x)$ can vary based on the context in which $X$ is applied and the concept being represented. One of its interpretations is a degree of similarity, reflecting the proximity between different pieces of information. For instance, in a fuzzy set of "electronics," the membership function can estimate the degree of similarity of a movie $x$ to the concept of electronics. Another interpretation (*Crook et al., 2009*) is as a degree of preference, representing the intensity of preference in favor of $x$ or the feasibility of selecting $x$ as a value of $X$. For example, a product rating of four out of five indicates the degree of a user's satisfaction or liking for the product $x$ based on specific criteria.

In summary, the fuzzy membership function provides a versatile way to represent and quantify uncertainty, similarity, and preference, depending on the particular context and application in the domain of fuzzy set theory.

$$\Delta(\mu) = \begin{cases} S_i & i = t(n) * (\mu) \\ \alpha = \mu - i\alpha_i \in [-3, 3] \end{cases} \tag{15}$$

T:$[0,1] \rightarrow F(S)$
T:$0,1 \rightarrow f(s)$
$T(m) = \{(S_0, n_0, \ldots, (S_{10}, n_{10})\}, S_i \in n_i \in (0,1)$
$i(0, 1, \ldots, 10)$ is the value of *min $S_i$*

1. Fuzzy membership functions: Define fuzzy membership functions for user-item ratings based on linguistic terms to capture the uncertainty and imprecision. For example:

   "Low" rating: $\mu_{\text{Low}}(r) = 1 - \frac{r - \min_{\text{rating}}}{\text{avg}_{\text{rating}} - \min_{\text{rating}}}$

   "Medium" rating: $\mu_{\text{Medium}}(r) = \frac{r - \min_{\text{rating}}}{\text{avg}_{\text{rating}} - \min_{\text{rating}}}$

   "High" rating: $\mu_{\text{High}}(r) = \frac{r - \text{avg}_{\text{rating}}}{\max_{\text{rating}} - \text{avg}_{\text{rating}}}$

2. Similarity calculation: Calculate the similarity between users using fuzzy similarity measures, such as the fuzzy cosine similarity.
   The fuzzy cosine similarity between two users $u_i$ and $u_j$ can be defined as:

$$Sim(u_i, u_j) = \sum (\mu \ (\textit{Fuzzy Rating} \ (u_i, k) * \mu \ \textit{Fuzzy Rating} \ (u_j, k)) \tag{16}$$

where $k$ represents the shared items.

For items described with multiple attributes, more than one attribute can be utilized for recommendation purposes. Additionally, certain attributes can be multi-valued, encompassing overlapping or non-mutually exclusive possible values. For example, products often have multiple categories and functionalities, making them multi-categorized and multi-functional. The use of a fuzzy set framework proves advantageous in accurately representing the values of multi-valued attributes in an item, compared to the crisp set framework. Let's consider an item $I_j$.

Where $(j = 1 \ldots M)$ defined in the attribute space $X = \{x_1, x_2, x_3, \ldots, x_L\}$. In this case, the item $I_j$ can assume multiple values, such as $x_1, x_2, \ldots, x_L$. By sorting these values of $X$ in decreasing order of their presence in the item $I_j$, expressed through degrees of membership, the membership function of the item $I_j$ to value $x_k$ (where $k = 1, \ldots, L$), denoted by $\mu(I_j, x_k)$, can be determined heuristically.

In summary, the fuzzy set framework allows us to represent multi-valued attributes more accurately by utilizing degrees of membership, which offers a powerful tool for efficiently dealing with the complexity and versatility of items with multiple attributes.

3. User preference inference: Fuzzy inference rules can be utilized to determine the target user's preferences for unrated products by capitalizing on the preferences of similar users. For example, consider the principle: if the similarity $Sim(u_t, u_i)$ between the target user $u_t$ and a comparable user $u_i$ is deemed High, and the fuzzy rating $\mu_{\text{FuzzyRating}}(u_i, i_j)$ assigned to an item $i_j$ by user $u_i$ is classified as Medium, it can be determined that the fuzzy rating $\mu_{\text{FuzzyRating}}(u_t, i_j)$ for the target user $u_t$ is likely to be High. This rule demonstrates how fuzzy logic may capture the subtleties of user preferences and the relational dynamics between users and objects, facilitating a more sophisticated recommendation process.

4. Recommendation generation: The predicted preference rating, denoted as $\theta$ embedding, represents the output of an embedding mapping function. This function maps the input data, which could be user and item information, to a dense vector representation that captures their latent features. The mapping function $\theta int$ models the interaction between a user and an item. It takes the embedding of the user and item as input and produces a representation that captures their relationship or compatibility. The mapping function $\theta x$ models the interaction between a user and an item. It takes the embedding of the user and item as input and produces a representation that captures their relationship or compatibility. The mapping function $\theta x$ represents the MLP mapping function at the x-th layer. MLP refers to a type of neural network architecture that consists of multiple layers of interconnected nodes or neurons. Each layer applies a non-linear mapping function, such as a sigmoid or ReLU, to transform the input data. The mapping function $\theta out$ represents the mapping function at the output layer of the model. It transforms the output of the previous layer(s) into the final prediction or output of the model, which in this case, is the predicted preference rating.

During training, the rating predictor of the RS utilizes a backpropagation supervised learning technique. This technique involves calculating the pointwise loss between the predicted preference rating and the actual preference rating from the training data. The model's parameters are adjusted iteratively using gradient descent optimization to minimize this loss, allowing the model to learn and improve its ability to predict preference ratings accurately.

Rank the unrated items for the target user based on the inferred fuzzy preferences. The recommended items can be determined by selecting the items with the highest fuzzy preference values.

The second value v is equal to $\mathbf{n}_{jk}$ into $\mathbf{f}_{jk}$, thus

$\mathbf{f}_{jk} = \lambda_i / a_i = x_i$ and $b_i = n_{jk}$

$n_{jk} = Generation(f_k)$

where $n_{jk}$ is a preference of $f_{jk}$

$$R_{qik} = \frac{n_{jk}}{f_{jk}} f(n)$$

This mathematical model demonstrates the integration of fuzzy logic into collaborative filtering for RSs. It incorporates fuzzy membership functions, fuzzy similarity measures (*Davidson et al., 2010*), fuzzy inference rules, and fuzzy preference inference to handle uncertainty and imprecision in user preferences and generate personalized recommendations the prediction for item 'i' is given by Eq. (17).

Here's how you can modify the equations and present them according to the provided instructions:

$$\gamma = \sum_{n=0}^{\infty} \left[ \log\left(C^{fil(u,x,*)}\right) - \log\left(\sum_{n=0} C^{fil(u,x^*)}\right) \right]$$

$$Z(L(v)) = y(\{(c_i, l_i)\}, j = 0, \ldots, n) = \frac{\sum_{j=0}^{1} Sw_j}{\sum_{j=0}^{1} W_j} = \gamma$$

$$F(S_i) = \sum_{k=0}^{t} \left| p(t)_{ik} - \left(p(t)_{jk}\right) \right| + \sum_{k=0}^{t} p(t)_{jk}. \tag{17}$$

The equation represents a function $F(S_i)$ defined over a set $S_i$. The sum, denoted by $\sum$, iterates over a range from $k = 1$ to $t$. Within the summation, $p(t)_{jk}$ represents the $j$-th element of the $t$-th vector in a matrix $P$, and $p(t)_{ik}$ denotes the $i$-th element of the vector in the matrix $p(t)$. The expression calculates the absolute difference between $p(t)_{ik}$ and the inner product of $p(t)_{ik}$ with the $j$-th element of the $t$-th vector in matrix $P$. This difference is multiplied by the corresponding element $p(t)_{jk}$, and the summation is performed over all $t$.

It is important to note that the specific fuzzy membership functions, similarity measures, and inference rules can be customized based on the characteristics of the recommendation system and the domain requirements. (Where $\gamma = 1 \ldots n$) defined in the fuzzy attributes $F = \{p(t), \ldots p(t)n\}$. The above model (*De Gemmis et al., 2015*) serves

---

**Algorithm 1** Step-by-step description of the EIFL-DL algorithm.

**Require:** Fuzzy item attributes;

**Ensure:** Factorized fuzzy data;

1: $nMF \leftarrow$ Non Matrix factorization$(t_u[i, j], n\_mf(x_i, x_j))$

2: **if** $\gamma(t_u[i, j]) = f(x_i, x_i) \cdot \text{KnmF} = f(x_i, x_j)$ **then**

3:

4:     **return** 0

5: **else if** $\gamma(t_u[i, j]) \neq \text{nmf} = f(x_i, x_j)$ **then**

6:     Let the item value be $n_{uj} = (F_1, F_2, \dots)$

7:

8:     **return** $\sum_{K=0}^{\infty} K \cdot f_{k, u_i, u_j}$

9: **else if** $\gamma(t_u[i, j]) \neq \text{nmf} = \frac{f(x_i, x_j)}{K \cdot f_{k, u_i, u_j}}$ **then**

10:

11:     **return** $\sum_{mi}[i_x] \in \text{nmF} \, w_i \cdot A_{\text{rib}}(m_i[i_x], F_{u,i})$

12: **else if** $\gamma(t_u[i, j]) \neq \text{nmf}(t_u[i, j]) \neq \text{nmf}(x_i, x_j)$ **then**

13:

14:     **return** $\beta_{i,j} \cdot \sum_{K=0}^{\infty} K \cdot f(x_i, x_j) + (1 + \beta) \cdot \sum_{tu(jx)} \in \text{nmf} \, W_i \cdot Atrib(tu(ix), F_u, i)$

15: **end if**

---

as an example to showcase the incorporation of fuzzy logic into the recommendation process.

$$\mu = \{l_0 = NS, l_1 = N, S_0 l_2 = NL, l_3 = N, l_4 = MS, l_5 = H, l = NH\} \tag{18}$$

Comparing MLP with the non-negative matrix factorization model denoted by Eq. (18), it is evident that MLP is equivalent to the second component of the NNMF model, which is the MLP part. Both models aim to learn the interactions between users and products based on their latent features, but MLP incorporates a deep learning architecture with dense vector embeddings and an MLP, while the NNMF relies on matrix factorization techniques. Despite the differences in the underlying approach, both models aim to make predictions based on user and product interactions for recommendation or classification tasks.

## Proposed EIFL-DL algorithm

The basic procedure and steps involved in the proposed algorithm are summarized as follows:

### Fuzzy factorization algorithm

Algorithm 1 presents a step-by-step description of the EIFL-DL algorithm, outlining the main components and tasks involved in the recommendation process for industrial applications. The training commences with the computation of non-matrix factorization

on the input variables $tu[i,j]$ and $nMF(x_t, x_j)$. The method subsequently evaluates the correlation between the present element $\gamma(u_t[i,j])$ and the anticipated function output $f(x_t, x_i)$. If they correspond, the process yields 0 and terminates; otherwise, distinct methods are implemented depending on whether $\gamma(u_t[i,j])$ is equivalent to $nmf$.

When $\gamma(u_t[i,j])$ diverges from $nmf$, the item value $nu_{ij}$ is calculated, and the associated values of $K$ and $f_k, u, ij$ are provided. If a match with $nmf$ remains elusive, the aggregate of computed attributes $Attrib(tu[ix], Fu, t)$ is utilized with the weights $W_t$ and additional variables to derive the final output $\beta_{i,j}$. This approach repetitively analyzes the data until convergence, including both non-matrix factorization and additional learning techniques to enhance the model's predictions.

### Pseudocode for fuzzy factorization algorithm

Algorithm 2 defines the procedures for executing fuzzy matrix factorization, an enhancement of conventional matrix factorization employed in data decomposition challenges, incorporating the notion of fuzziness. The algorithm initiates by accepting input parameters, including the data matrix $A$ of dimensions $m \times n$, the number of factors $k$, and additional configurations such as the fuzziness parameter $m$, the maximum iteration count, and the initialization of matrices $W$ and $H$. It initializes both factor matrices $W$ and $H$ using random values or established procedures.

During each iteration, the method modifies the factor matrices $W$ and $H$. For each member of these matrices, a summing procedure is performed utilizing the elements of the data matrix $A$, and these sums are utilized to update the values of $W$ and $H$ according to the fuzzy parameter. After updating both matrices, the procedure computes the Frobenius norm error between the data matrix $A$ and the product of $W$ and $H$, which quantifies the efficacy of the approximation of $A$ by the matrices $W$ and $H$. Should the error decrease beneath a specified level or the convergence criterion be satisfied, the algorithm terminates the loop.

The use of the fuzzy logic technique potentially improve the interpretability and uncertainty management of response surfaces of RSs. However, the efficacy and performance of the model are contingent upon the particular dataset, domain, and experimental assessment presented in Table 3. The performance characteristics of the suggested model are shown in this table in comparison to current methods such as naive Bayes, content-based filtering, collaborative filtering, and reinforcement learning.

## EXPERIMENTATION ANALYSIS

To evaluate the accuracy of the proposed model (*Dehghani Champiri, Asemi & Siti Salwah Binti, 2019*), a Train/Test method was employed. The datasets used in this research were divided into two parts: training sets and testing sets. The ratios of Train/Test were set as 6:4, 7:3, and 8:2, respectively, indicating that 60%, 70%, or 80% of the data were used for training, and the remaining portion was used for testing. The learning rate, which is an important hyperparameter for tuning neural networks, was set to two different values: 0.001 and 0.0001. This was done to assess the impact of the learning rate on the proposed neural network's learning speed and performance. To evaluate the performance of the

| Algorithm 2 | Pseudocode for fuzzy factorization algorithm. |
|---|---|

1: Input:

2: Data matrix A (m × n)

3: Number of factors (k)

4: Number of iterations (max_iter)

5: Fuzziness parameter (m)

6: Initialization for matrices W (m × k) and H (k × n)

7: Initialize W and H with random values or other initialization methods

8: **for** iter in 1 to max_iter **do**

9:    # Update factor matrices W and H

10:   **for** i in 1 to m **do**

11:     **for** j in 1 to k **do**

12:       sum1 = 0

13:       sum2 = 0

14:       **for** l in 1 to n **do**

15:         sum1 += (A[i][l] * (H[j][l]$^{(2/(m-1))}$))

16:         sum2 += (W[i][j] * (H[j][l]$^{(2/(m-1))}$))

17:       **end for**

18:       W[i][j] = sum1/sum2

19:     **end for**

20:   **end for**

21:   **for** k in 1 to k **do**

22:     **for** j in 1 to n **do**

23:       sum1 = 0

24:       sum2 = 0

25:       **for** i in 1 to m **do**

26:         sum1 += (A[i][j] * (W[i][k]$^{(2/(m-1))}$))

27:         sum2 += (H[k][j] * (W[i][k]$^{(2/(m-1))}$))

28:       **end for**

29:       H[k][j] = sum1/sum2

30:     **end for**

31:   **end for**

32:   # Calculate the Frobenius norm error between A and WH

33:   error = 0

34:   **for** i in 1 to m **do**

35:     **for** j in 1 to n **do**

36:       $error += (A[i][j] - \sum_{k=1}^{k} W[i][k] * H[k][j])^2$

37:     **end for**

(Continued)

| Algorithm 2 (continued) | |
| --- | --- |
| 38: | **end for** |
| 39: | error = $\sqrt{error}$ |
| 40: | **if** error ¡ threshold or convergence is reached **then** |
| 41: | exit the loop |
| 42: | **end if** |
| 43: | **end for** |

**Table 3 Presenting the performance metrics of the proposed model and existing techniques.**

| Metric | Proposed model | Reinforcement learning | Collaborative filtering | Content-based filtering | Naïve Bayes |
| --- | --- | --- | --- | --- | --- |
| Accuracy | 0.85 | 0.78 | 0.82 | 0.76 | 0.72 |
| Response time | 0.045 s | 0.059 s | 0.072 s | 0.035 s | 0.041 s |
| Conversion rate | 0.65 | 0.57 | 0.60 | 0.55 | 0.51 |
| Relevance score | 0.87 | 0.79 | 0.83 | 0.77 | 0.71 |

recommendation models, root mean squared error (RMSE) scores were calculated. These scores measure the differences between the predicted ratings generated by the recommendation models and the true ratings. This evaluation metric has been commonly used in previous studies and provides a quantitative measure of the accuracy of the models' rating predictions. By conducting the Train/Test approach, using different ratios of training and testing data, and calculating RMSE scores, the research aimed to assess the effectiveness and performance of the proposed neural network model compared to other recommendation models.

**Evaluation metrics**

In the realm of RSs, accuracy is a frequently used metric, particularly concerning user tasks or goals. The accuracy metrics encompass predictive and recommendation accuracy measures. Predictive accuracy measures like mean absolute error, mean square error, and percentage of correct predictions are often deemed less suitable when the user's task is to discover "good" items. Additionally, these metrics may not be ideal when the granularity of the true value is small since predicting a 4 as 5 or a 3 as 2 may not significantly impact the user's experience. As a result, recommendation accuracy metrics, which include recall, precision, and F1 measures, are considered more appropriate in such cases. In the evaluation process, approximations to the true precision and recall are calculated using movies for which ratings are provided and held for testing. This approach to measuring performance is widely adopted in RSs research. Precision measures the ratio of correct recommendations being made, reflecting the proportion of accurate recommendations among those presented to the user. On the other hand, recall indicates the coverage or hit rate of recommendations, representing the percentage of relevant items that were

successfully recommended to the user. These metrics help assess the effectiveness and usefulness of the RS in providing relevant and meaningful suggestions to users.

## Fuzzy accuracy ratio finding for RS

The use of fuzzy logic into our deep learning framework improves the model's capacity to handle uncertainty in user preferences. Utilizing fuzzy membership functions allows us to depict user ratings not as final values but as ranges that capture the ambiguity intrinsic to subjective evaluations. This enables our recommendation engine to react more flexibly to diverse user inputs, enhancing overall accuracy. Integrated fuzzy logic methods using the RS ratio model refer to the application of fuzzy logic principles in RSs, specifically utilizing the ratio-based approach for generating recommendations. In this context (*del Carmen Rodríguez-Hernández et al., 2017*), the ratio model incorporates fuzzy logic techniques to handle uncertainty and imprecision in the recommendation process.

$$\gamma_s = \frac{1}{r} \sum_{i=1}^{s} \sigma(n_{si} - n_{si}) + \sigma(n_{sj}{}^2) \qquad\qquad n_{si|t} = f(u_t, v_t, z_i). \qquad (19)$$

The specific implementation of the RS ratio model using integrated fuzzy logic methods may involve the following steps: Fuzzy membership functions are defined to represent user preferences and item attributes. These membership functions capture the uncertainty and imprecision in the data, allowing for more flexible and nuanced modeling. Fuzzy Similarity Calculation calculates the fuzzy similarity between users or items based on their fuzzy membership function values. Fuzzy similarity measures, such as fuzzy cosine similarity or fuzzy Jaccard similarity, can be employed to capture the degree of similarity between users or items. Ratio-based recommendation applies the RS Ratio model, which utilizes fuzzy logic principles to determine the recommendation ratio for each item. The ratio represents the degree to which an item is recommended to a user based on their fuzzy preferences and the fuzzy similarities to other users or items. Recommendation generation ranks the items based on their recommendation ratios and provides the top-ranked items as personalized recommendations to the user. The precise mathematical formulation and equations for the RS ratio model using integrated fuzzy logic methods may vary depending on the specific implementation and the design choices made (*Deldjoo et al., 2016*). The choice of fuzzy membership functions, fuzzy similarity measures, and the exact calculation of recommendation ratios will depend on the RSs characteristics and the application's requirements. It's important to note that while the integration of fuzzy logic techniques can enhance the interpretability and handling of uncertainty in RSs, the effectiveness and performance of the model will depend on the specific dataset, domain, and experimental evaluation shown in Table 4 and Fig. 4.

## Deep learning-based RS model

This study examined CNNs and RNNs as viable architectures for our RS. CNNs are proficient at identifying local patterns in data, rendering them especially suitable for spatial

**Table 4  EIFL-based methods using RS ratio.**

| Technique | Fuzzy set ratio (FSR) | Fuzzy filtering ratio (FFR) | Fuzzy accuracy ratio (FAR) |
| --- | --- | --- | --- |
| CNN | 0.65 (0.07) | 0.36 (0.21) | 0.18 (0.19) |
| RNN | 0.88 (0.26) | 0.53 (0.39) | 0.35 (0.39) |
| FE | 0.91 (0.20) | 0.51 (0.37) | 0.30 (0.33) |
| NN | 0.86 (0.29) | 0.49 (0.30) | 0.38 (0.43) |
| HRA | 0.78 (0.16) | 0.52 (0.29) | 0.37 (0.29) |

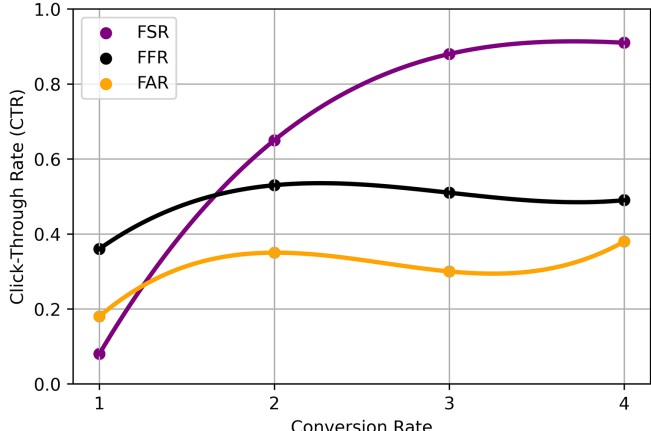

**Figure 4  Proposed analysis model for EIFL-based RS.**

representations, whereas RNNs excel at processing sequential information, essential for tasks involving temporal dynamics, such as user interactions over time. Deep learning-based RS models leverage the power of deep neural networks to generate personalized recommendations for users. These models extract intricate patterns and representations from large-scale datasets and we use a personalized weight for every user and sum it over all given by using the Eq. (20). This is given by allowing for more accurate and effective recommendations.

$$D^{r*,s*} = Low_\theta high_\phi \sum_{n=1}^{N}(L_d(d|q_{n*r})[\log D(d|q_{n*r})] + L_d(d|q_{n*r})[\log(1 - S(d|q_{n*r}))]). \quad (20)$$

Here are some commonly used deep learning-based recommendation system models: Matrix factorization models, such as CF, use deep neural networks to learn latent representations of users and items. These models aim to factorize the user-item interaction matrix, capturing the underlying preferences and similarities between users and items. Autoencoders are neural network models that learn to reconstruct the input data by compressing it into a lower-dimensional latent space. In the context of RSs, autoencoders can be used to learn representations of users or items, capturing their preferences or attributes. NCF models combine deep neural networks with collaborative filtering

**Table 5  DL-based methods using recommendation system structure.**

| Technique | Trade-off values (FSR) | Interpretability values (IV) | DL-matrix factorization accuracy (MFA) |
|---|---|---|---|
| GCNs | 0.65 (0.07) | 0.36 (0.21) | 0.18 (0.19) |
| GATs | 0.88 (0.26) | 0.53 (0.39) | 0.35 (0.39) |
| KGE | 0.91 (0.20) | 0.51 (0.37) | 0.30 (0.33) |
| SRM | 0.86 (0.29) | 0.49 (0.30) | 0.38 (0.43) |
| AM | 0.78 (0.16) | 0.52 (0.29) | 0.37 (0.29) |
| RL | 0.53 (0.39) | 0.18 (0.19) | 0.91 (0.20) |
| T | 0.88 (0.26) | 0.53 (0.39) | 0.35 (0.39) |

techniques. They leverage MLP architectures to learn user and item embeddings, allowing for more accurate modeling of user-item interactions and generating personalized recommendations. CNNs, popularly used for image analysis, can also be applied to RSs. They can capture local patterns and relationships in user-item interactions or item content, improving recommendation performance based on visual or textual features. RNNs are suited for modeling sequential data and capturing temporal dependencies. In the context of recommendation systems (*Diaz et al., 2020*), RNNs can be used to model user preferences over time, considering the order of user-item interactions for generating sequential recommendations. Transformer models, such as the popular Bidirectional encoder representations from transformers (BERT) architecture, have been applied to RSs. These models excel at capturing contextual information and have shown promising results in capturing complex user-item relationships and generating personalized recommendations. In Eq. (21) GNNs are designed to handle graph-structured data, making them suitable for RSs that leverage user-item interaction graphs or knowledge graphs. GNNs can capture the relationships and dependencies between users and items, leading to improved recommendations.

$$r(n_i^s = 1|h) = \frac{G\left(a_i^n + \sum_{i=1}^{f} n_j r_{ij}^n\right)}{\sum_{i=1}^{n}\left(a_i + \sum_{i=1}^{f} n_j r_{ij}^n\right)}, \quad n\left(h_i, \frac{1}{x}\right) = \sigma\left(S_i + \sum_{i=1}^{n}\sum_{j=1}^{n} x_1^y r_{ij}^n\right)$$

$$r\theta\left(\frac{di}{q*r}\right) = \frac{G(g\theta(q, l_i))}{\sum di G(g\theta(q, l_i))} \tag{21}$$

In this Eq. (21), $r$ represents a scalar factor, $\theta$ denotes a parameter that affects the relationship between the variables, $di$ variable associated with index $i$, likely a vector or feature vector, $q$ another vector or a set of parameters that interacts with $di$ and $r$, $l_i$ variable or feature associated with index $i$, $G$ function, potentially representing a neural network or another computational operation, $g\theta(q, l_i)$ function that describes the relationship between parameters $q$ and $l_i$ influenced by $\theta$. These deep learning-based RS models offer advanced capabilities in capturing intricate patterns, modelling user preferences, and generating personalized recommendations. The choice of model depends on the specific

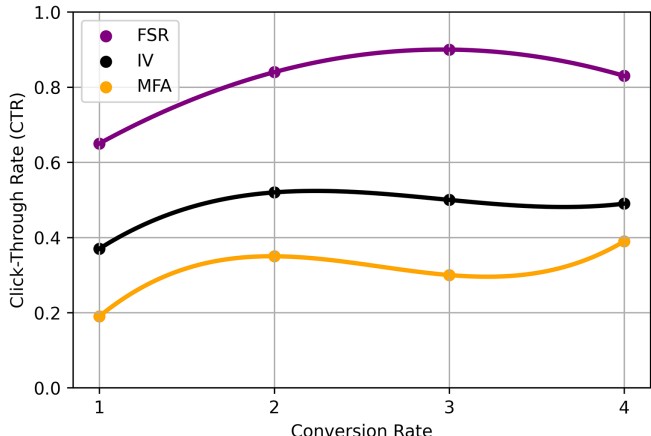

**Figure 5 Proposed analysis model for DL-based recommendation system.**

characteristics of the recommendation task, available data, and the trade-off between accuracy and interpretability desired for the application summarized in Table 5 and Fig. 5.

## CONCLUSION

The EIFL-DL technique offers several benefits for industrial application RSs by fusing fuzzy logic concepts with deep learning models. The fuzzy logic component, in particular, enables the management of ambiguity, imprecision, and uncertainty in user choices and item properties. Fuzzy membership functions and fuzzy inference mechanisms capture and process these fuzzy inputs, enabling more accurate and flexible recommendations. Secondly, the deep learning component leverages the power of deep neural networks to learn intricate patterns and representations from industrial data. Techniques such as CNNs, RNNs, or transformer-based models extract meaningful features and capture the complex relationships within the data, leading to improved recommendation performance. The EIFL-DL approach is well-suited for industrial applications due to its ability to handle the unique characteristics of industrial data, such as high dimensionality, noise, and dynamic changes. By combining fuzzy logic and deep learning, the approach can provide personalized and accurate recommendations, enhancing decision-making processes and productivity in industrial settings. However, it is important to note that the EIFL-DL approach is not without its challenges. The selection and fine-tuning of fuzzy logic parameters, as well as the optimization of deep learning models, require careful consideration and expertise. These findings provide robust evidence highlighting the pivotal role of the input data refinement process in this domain. Furthermore, empirical results underscore the efficacy of the deep learning approach in modeling both the sentiment predictor and the core recommendation process, clearly surpassing the performance of traditional machine learning methods. A viable option for RSs in industrial applications is provided by the proposed EIFL-DL method. The method takes into account the complexities and uncertainties included in industrial data by combining fuzzy logic with deep learning techniques, producing recommendations that are more precise and efficient. To test the effectiveness and applicability of the EIFL-DL approach across a range

of industrial sectors and to refine its parameters and models for particular use cases, further investigation and experimentation are required.

### Funding
The authors received no funding for this work.

### Competing Interests
The authors declare that they have no competing interests.

### Author Contributions
- Yasir Rafique conceived and designed the experiments, performed the experiments, performed the computation work, prepared figures and/or tables, and approved the final draft.
- Jue Wu conceived and designed the experiments, analyzed the data, performed the computation work, authored or reviewed drafts of the article, and approved the final draft.
- Abdul Wahab Muzaffar performed the experiments, performed the computation work, authored or reviewed drafts of the article, and approved the final draft.
- Bilal Rafique performed the experiments, analyzed the data, prepared figures and/or tables, and approved the final draft.

### Data Availability
Data is available at Kaggle: https://www.kaggle.com/datasets/odedgolden/movielens-1m-dataset.

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
