# Peer review of "An enhanced integrated fuzzy logic-based deep learning techniques (EIFL-DL) for the recommendation system on industrial applications"

_PeerJ Computer Science, doi:10.7717/peerj-cs.2529_

## Round 0.1 · original submission · Major Revisions

The reviewers noted issues with the quality of figures and formulas, recommending improvements in clarity, reduction of redundancy, and consistent indentation. Furthermore, the lack of explanations for numerous formulas and algorithms in the paper raised concerns about reader comprehension. The reviewers also suggested comparing the proposed method with commonly used techniques in the field for a more comprehensive evaluation. While the paper reports high accuracy and other metrics for the proposed method, there was a request for a more significant baseline comparison and a detailed description of evaluation results. Concerns were raised about the ad-hoc nature of pattern retrieval using deep learning and the complexity of fine-tuning fuzzy logic parameters and deep learning models. Addressing these issues and incorporating the recommendations would enhance the research's quality.

**Language Note:** The review process has identified that the English language must be improved. PeerJ can provide language editing services - please contact us at [email protected] for pricing (be sure to provide your manuscript number and title). Alternatively, you should make your own arrangements to improve the language quality and provide details in your response letter. – PeerJ Staff

Reviewer 1 ·

Basic reporting

In summary, the research introduces an Enhanced Integrated Fuzzy Logic-based Deep Learning Technique (EIFL-DL) for industrial recommendation systems. The experimental design needs clearer organization, suggesting smaller subsections and visual aids for better understanding. Strengthening the validity of findings involves providing a detailed description of EIFL-DL methodology, explaining the integration of fuzzy logic and deep learning, and addressing potential limitations. Attention is needed for clear appendix images, refined writing style, and detailed annotations for mathematical formulas. By addressing these points, the research can enhance its contribution to industrial recommendation systems.

Experimental design

Please refine the logic and organization of the experimental design section for better clarity. Break down the design into smaller, well-defined subsections, outlining variables, controls, and procedures for each experiment. Incorporate visual aids like flowcharts to illustrate the experimental process. Clearly explain the rationale behind experimental setups and consider providing a concise summary of conditions, data sources, and variations.

Validity of the findings

While the optimization of the recommender system using deep learning and fuzzy learning is reasonable, the description of the system's methodology in this paper is unclear, and the logic is fuzzy.
Please enhance the validity of the findings by providing a detailed description of the EIFL-DL methodology. Clearly articulate the integration of fuzzy logic and deep learning, emphasizing how it addresses challenges in industrial data. Discuss the choice of deep learning techniques, such as CNNs and RNNs, in feature extraction, and explain their contribution to recommendation accuracy. Address potential limitations, perform a thorough validation, and present statistical measures or benchmarks for credibility.

Additional comments

1. Appendices: Revise the images in the appendices to eliminate blurriness and distortion. Ensure that all visual elements are clear, enhancing the overall readability of supplementary materials.

2. Writing Style: Refine the writing style by minimizing redundancy and repetition. Clearly delineate specific terms and avoid vague references. Ensure that each section contributes uniquely to the overall narrative without unnecessary repetition.

3. Mathematical Formulas: Provide detailed annotations and explanations for all mathematical formulas. Clearly define symbols and subscripts, leaving no room for ambiguity. This step is crucial for readers to comprehend the mathematical intricacies of the proposed EIFL-DL framework.

4. Section 3.1.1: The reference to "dataset 1" is abrupt and unclear. Explain the reference to "dataset 1" more explicitly. Clarify its relevance and integration within the larger context of the research.

5. Section 3.2.1: The motivation and specific methodology are unclear. An explanation of each step's purpose and results is needed. Provide a more comprehensive explanation of the motivation behind Section 3.2.1, detailing the objectives and expected outcomes. Additionally, offer a clearer and more detailed description of the specific steps involved in this section, enhancing reader understanding.

6. Paragraphs in the article inconsistently exhibit indentation. Please ensure uniform indentation throughout the document.

7. The logic in the experimental design is unclear, with a dense accumulation of experimental information, making it challenging to differentiate between various cases. It is recommended that the author divides the experiments into smaller sections, providing clear descriptions of each experiment's conditions, results, and an analysis of the outcomes.

Typo:
Data..->data. (Abstract)
Data->data (Line 193)

Reviewer 2 ·

Basic reporting

This paper proposes an Enhanced Integrated Fuzzy Logic-Based Deep Learning Techniques (EIFL-DL) framework for recommendation systems in industrial applications. The authors aim to generate more accurate and interpretable recommendations for industrial applications by integrating fuzzy logic with deep learning. The fuzzy logic component, in particular, enables management of ambiguity, imprecision, and uncertainty in user choices and item properties. And the deep learning component leverages the power of deep neural networks to learn intricate patterns and representations from the industrial data. The paper evaluates the effectiveness using the Train/Test split method on the dataset. The authors evaluate the fuzzy accuracy ratio finding and deep learning based recommendation systems separately with different deep learning models. The result shows that the proposed method has high accuracy, conversion rate, relevance score and lower response time than the other existing techniques.

However, there are some issues about the paper reporting which may hinder the understanding of the readers of this paper. Firstly, the text in the figure is blurred especially in figure 1. which is hard to read and understand. Secondly, there are a large amount of formulas in this paper but there’s no explanation of any of them. Even with domain knowledge, the readers will be confused about the meaning of the variables in the formulas listed without context and explanation. It is the same for the algorithm part. It would be better to have text to describe each line of the algorithm.

Experimental design

The experiment is conducted on a dataset with 60,000 records with 5 labels. To improve the quality of the dataset, the system utilizes different deep learning models including GANs, VAEs, LSTMs in the different stages of the data processing pipeline. Also, deep learning based recommendation techniques are applied. Apart from the deep learning component, integrated fuzzy logic based techniques, which consist of fuzzy membership functions, similarity calculation, user preference inference and recommendation generation, are applied to fit in the industrial application setting to handle uncertainty and imprecision.

However, the technique part lacks some details. For example, the paper mentions that auto encoders and GANs can be used to remove noisy data. How does it work in the proposed system ? What’s the loss function and how are they trained ? The authors don’t demonstrate clearly. What’s more, there are only about 12 lines for deep learning based recommendation techniques. It would be better if the authors could list some commonly used methods and make simple comparisons.

Validity of the findings

The result shows that the proposed method has high accuracy, conversion rate, relevance score and lower response time than the other existing techniques including reinforcement learning, collaborative learning filtering, content-based filtering and naive bayes. Also, the paper makes a comparison of the performance of different techniques using metrics like fuzzy set ratio, fuzzing filtering ratio and fuzzy accuracy ratio. The dl-based methods are also evaluated.

Additional comments

I strongly recommend the authors to do multiple rounds of proofreading in order to avoid some confusing mistakes in writing.

Example


line 193-195. After “address two key challenges:”, there challenges are listed with (a)(b)(c)

line 248 After reading the sentence “cross-validation may be employed to assess the generalization capability of the model.”, I feel that the authors are not quite sure what method they are using for the evaluation.

Reviewer 3 ·

Basic reporting

The copy right format seems unnatural.
Most of the formulation requires a clearer definition with explanation.

Experimental design

The EIFL-DL technique combines fuzzy logic with deep learning. The fuzzy logic component handles ambiguity, imprecision, and uncertainty in user preferences and item characteristics. This is achieved through fuzzy membership functions and fuzzy inference mechanisms. The deep learning part utilizes neural networks like CNNs, RNNs, or transformer-based models to learn intricate patterns in industrial data. This helps in extracting meaningful features and understanding complex relationships within the data.

Here is some limitation regarding this approach:
- The pattern retrieval by deep learning based approach is somehow too ad-hoc for a seriously proposed algorithms. The authors fail to provide a specific reason why some approach should be chosen and why not, undermining the interpretability of such mechanism
- Selecting and fine-tuning the fuzzy logic parameters and optimizing deep learning models require significant expertise and careful consideration. This complexity can be a barrier to implementation and may lead to inefficiencies if not handled correctly.
- The integration of deep learning models often requires substantial computational resources. This can be a constraint for smaller industries or applications where such resources are limited.

Validity of the findings

The evaluation for the Fuzzy Accuracy ratio in recommendation system is reported but a more significant baseline is missing.
The detailed description of the evaluation result is required for the readers to better understand what lev el of improvement is achieved with this Fuzzy logic.

---

## Round 0.2 · Minor Revisions

The authors have made significant improvements to the manuscript, particularly in enhancing language clarity, organizing figures, and refining formulas. while the integration of fuzzy logic and deep learning is better explained, a more in-depth discussion of the differences between CNNs and RNNs and their impact on accuracy, supported by evaluation results, would strengthen the validity of the findings.

Reviewer 1 ·

Basic reporting

The language has improved, though minor issues still remain. Figures and formulas are now clearer, and the explanations are more comprehensive. The paper is well-referenced and follows academic standards.

Experimental design

The authors provided a more detailed explanation of the experimental design, breaking it down into clearer subsections. They addressed the comments regarding the complexity of the experimental setup by incorporating visual aids such as flowcharts and providing descriptions of each step in their experiment. This restructuring has improved readability and comprehensibility. However, the logic behind some methodological choices (such as the use of certain deep learning models) could still be explained more rigorously

Validity of the findings

The integration of fuzzy logic and deep learning is now better explained. Additional baseline comparisons have strengthened the validity of the results.

Additional comments

The authors addressed most concerns, improving figure quality, clarifying formulas, and organizing the experimental section better.

Reviewer 4 ·

Basic reporting

Since this is a resubmission that includes rebuttal notes, my focus will primarily be on whether the authors have addressed the previous reviewers' comments.

Previous reviewers recommended improving the clarity of formulas, and I would suggest the same. For instance, the authors could add a table summarizing the variable notations used in the paper and their meanings. While some notation explanations are provided in the text, such as for equations 1, 2, and 4, a table would offer a clearer summary. Additionally, certain notations are missing explanations, such as "alpha" and "d_j" in equation 3, and "gamma" in equation 9. I recommend that the authors review all equations to ensure that every notation is properly explained.

The authors have included pseudocode to explain the algorithm, which is a positive addition. However, I would also suggest adding a paragraph that explains the pseudocode in detail, outlining how the code functions, including specific sections like what lines 10-20 are doing.

Experimental design

Previous reviewers highlighted shortcomings in the experimental design, such as the need to break down the design and explain the rationale behind experimental setups. While the authors provided some explanation in the rebuttal, I do not see this incorporated into the diff file. I suggest integrating the explanations from the rebuttal into the paper to help readers better understand the experimental design.

One reviewer raised the question: "The paper mentions that autoencoders and GANs can be used to remove noisy data. How does this work in the proposed system? What is the loss function, and how are they trained?" However, there is no response to this in either the rebuttal or the paper. I would recommend that the authors add some text to clarify this aspect.

Validity of the findings

Reviewers also mentioned the need to "articulate the integration of fuzzy logic and deep learning" and to "discuss the choice of deep learning techniques." While the authors provide some explanation in the rebuttal, it lacks concreteness. For example, what are the differences between CNNs and RNNs in this context, and how would accuracy differ? It would be helpful if the authors could provide evaluation results to support these points.

---

## Round 0.3 · accepted · Accept

The authors have enhanced clarity in language and experimental details, presenting content more coherently. Congratulations!

Reviewer 1 ·

Basic reporting

The revisions made to this article are satisfactory overall. The authors have enhanced clarity in language and experimental details, presenting content more coherently. Figures and references are well-structured, and the experimental design and interpretation of results are sufficiently comprehensive. The revisions strengthen the study’s innovation and applicability. A brief discussion on future research directions is recommended to further improve the article’s completeness.

Experimental design

N/A

Validity of the findings

N/A

Additional comments

N/A